# Optimal Policy Minimum Bayesian Risk

## Abstract

Inference scaling helps LLMs solve complex reasoning problems through extended runtime computation. On top of long chain-of-thought (long-CoT) models, purely inference-time techniques such as best-of-N (BoN) sampling, majority voting, or more generally, minimum Bayes risk decoding (MBRD), can further improve LLM accuracy by generating multiple candidate solutions and aggregating over them. These methods typically leverage additional signals in the form of reward models and risk/similarity functions that compare generated samples, e.g., exact match in some normalized space or standard similarity metrics such as Rouge. Here we present a novel method for incorporating reward and risk/similarity signals into MBRD. Based on the concept of optimal policy in KL-controlled reinforcement learning, our framework provides a simple and well-defined mechanism for leveraging such signals, offering several advantages over traditional inference-time methods: higher robustness, improved accuracy, and well-understood asymptotic behavior. In addition, it allows for the development of a sample-efficient variant of MBRD that can adjust the number of samples to generate according to the difficulty of the problem, without relying on majority vote counts. We empirically demonstrate the advantages of our approach on math (MATH-500) and coding (HumanEval) tasks using recent open-source models. We also present a comprehensive analysis of its accuracy-compute trade-offs.

## 1 Introduction

Recent progress in large language model (LLM) technologies has reignited interest in decoding methods, and in general in scaling laws for inference time compute. Reasoning models, such as OpenAI's O1, O3 and O4-mini (Jaech et al., 2024; OpenAI, 2025), Alibaba's Qwen with Questions[1] and DeepSeek's R1 (Guo et al., 2025), can learn to produce long chains of thought (long-CoT) that solve very hard problems by using reinforcement learning with verifiable rewards. At the same time, there is a general resurgence of interest in decoding methods beyond simple greedy decoding or sampling, see the recent NeurIPS tutorial (Welleck et al., 2024). Among these are methods that rely on complex tree traversal, such as Monte Carlo tree search (Browne et al., 2012; Chen et al., 2024) and different variants of beam search. A second category includes best-of-N (BoN) decoding and self-ensembling techniques that can exploit additional signal such as process reward models (Uesato et al., 2022; Lightman et al., 2024) or some measure of consistency across multiple model outputs – these can be generally regarded as variants of minimum Bayesian risk decoding (MBRD) (Kumar & Byrne, 2004; Bertsch et al., 2023).

A large body of recent research has focused on long-CoT inference scaling. Monte Carlo tree search has also received significant attention due to its success in other machine learning areas such as games. In this paper, we focus on the remaining category of BoN/MBRD methods, which offer a unique and complementary set of advantages over other approaches. First, these methods can be used with any model, irrespective of whether the latter was trained for long-CoT reasoning. Second, their implementation is relatively simple, often requiring only parallel decoding and a final integration of results, which gives more precise control over inference scaling budget. Finally, they are modular, allowing for the use of off-the-shelf reward models to support multiple domains, even beyond those where long-CoT is particularly advantageous. On the flip side, they typically provide a lower gain in accuracy for the same amount of compute[2]. Whether BoN or MBRD is the supe-

---

[1] https://qwenlm.github.io/blog/qwq-32b-preview/
[2] Early DeepSeek-R1 release note https://api-docs.deepseek.com/news/news1120.

rior option in a given scenario is dependent on the quality of the generator relative to that of the reward model. Finally, the simplicity of BoN and MBRD leaves less room for modification and improvement.

In this work, we propose an enhancement to standard MBRD, termed Optimal Policy MBRD (OP-MBRD), with the following desirable properties:

- Robust response and performance across different scenarios, outperforming or closely matching the better of BoN and MBRD, even when there is a large performance gap between the two.
- Well-understood asymptotic behavior, converging to regular MBRD over a distribution that balances the contributions of a reward model and a reference generator.
- A sample-efficient version of the algorithm that can adjust the number of generated samples depending on the difficulty of the prompt, relying on general string matching instead of exact match counts.
- OP-MBRD retains most of the simplicity of BoN and MBRD, while introducing only a single new parameter and remaining compatible with general MBRD (i.e., beyond the use of simple exact match as in ordinary majority voting).

## 2    RELATED WORK

The following approaches can be considered related to our work.

**Learned Chain-of-Thought Techniques:** These inference scaling methods achieve high performance by training the generator to produce chain of thought to solve difficult problems. These CoTs often contain spontaneously appearing instances of self-reflection, backtracking, option enumeration, summarization and others. The most successful models are trained with reinforcement learning and verifiable rewards (Jaech et al., 2024; OpenAI, 2025; Guo et al., 2025). There are however more structured approaches that consider specific types of skills in the CoT (Kumar et al., 2024; Gandhi et al., 2025). Compared with the approach presented here, these techniques require specific generator training but no special decoding, besides support for long context. They can be considered complementary to the technique presented here.

**Majority Voting and Bayesian Risk:** This concerns inference scaling methods that generate multiple outputs from a model and consolidate them into a final answer. Their main advantage is their simplicity, requiring only $N$ independent generations, no specific generator training, or need of an external model such as reward models. Recent variants applied to LLMs include self-consistency (Wang et al., 2023), and conventional counts-based consensus in mathematical reasoning (Yang et al., 2024; Guo et al., 2025). Minimum Bayes risk decoding (Kumar & Byrne, 2004) can be considered a generalization of majority voting (MV) that utilizes a risk or similarity function between pairs of outputs to select the output that has the lowest risk / highest similarity with regard to any other output. MBRD reduces to count-based MV by using exact match as similarity (Bertsch et al., 2023). MBRD allows to extend MV to domains where exact match is not an option. A disadvantage of MV methods is that they often require large amount of samples to yield good gains.

**Reward-weighted Post-Processing:** These can be seen as an enhancement of the previous. They generate $N$ independent sentences in the same way, but utilize a separate model to score the completed outputs e.g. a reward model. Best-of-N (BoN) (Charniak & Johnson, 2005) is a very common and simple method with proven success to enhance performance (Yang et al., 2024). These methods also include combinations of majority voting with reward models, which provide improved performance with respect to plain majority voting, e.g., voting verifiers (Li et al., 2023). These methods can also be expressed as a form of MBRD where pair-wise risk/similarity is enhanced with the output reward. Reward post-processing compensates some of the limitations of pure MV while adding only the overhead of a call to to an external reward model. The method here presented falls into this category. As shown in the next sections it retains the simplicity advantage of similar counterparts such as BoN or voting verifiers, while being derived from well understood principles, providing more robust performance and an efficient version with better performance/compute trade-offs.

**Step-by-Step Decoding:** This includes methods that generate $N$ outputs in steps. After each step all partial completions are scored and combined together to produce the prefixes for the next step.

This can be done through deterministic pruning of the worse options as in beam search (Graves, 2012), or stochastic re-sampling of candidates as in (Deng & Raffel, 2023). Scorers can be reward models (Deng & Raffel, 2023), but also attribute classifiers (Yang & Klein, 2021). With the rise of reasoning LLMs and process reward models (PRMs) (Lightman et al., 2024), able to score partial reasoning chains, steps have grown fro single tokens to multiple, although the basic results are maintained [3]. Compared to reward-weighted post-processing techniques, step-by-step additional complexity due to the need to synchronize intermediate steps and the extra communication overhead per step between generator and scorer. The technique introduced here could also be applied however to multi-step algorithms, but this is beyond the current scope of the manuscript.

**Efficient Inference Scaling** Optimal allocation of inference compute can enable inference-time methods to outperform simply using a larger model Snell et al. (2024). Given the recent success of inference scaling methods, several approaches have been proposed in this area. One category of methods estimates task difficulty and performs budget allocation or input routing to different generators (Damani et al., 2024). Other approaches focus on minimizing the number of samples generated or compared, based on the observed distribution of answers over multiple samples (Aggarwal et al., 2023) or pair-wise risk (Cheng & Vlachos, 2023).

**Optimal Policy Approaches** The method introduced here is also related in its mathematical background to recent works using Optimal Policy. Methods like proximal policy optimization (PPO) (Schulman et al., 2017), Direct preference optimization (DPO) (Rafailov et al., 2023), GDC++ (Korbak et al., 2022) and BRAIn (Pandey et al., 2024) utilize the same KL-controlled reward maximization objective but for the purpose of deriving a Reinforcement Learning loss. Rejection Sampling Optimization (RSO) (Liu et al., 2024) modifies DPO to use samples from the optimal posterior via rejection sampling, while Guide Speculative Inference (GSI) Geuter et al. (2025) leverages it for speculative decoding in inference scaling. The former is also a RL learning algorithm while the later is related to our use of log-ratio, see experimenta setup.

## 3 Minimum Bayesian Risk Decoding

We define an LLM as a neural network parameterizing a distribution $p(y \mid x)$ over strings. Here $x, y \in V^+$ are input and output strings, respectively, and $V^+$ is the countably infinite set of all possible strings formed by concatenating tokens from a vocabulary $V$. Generating from an LLM generally corresponds to finding the most likely string

$$\hat{y} = \arg \max_{y \in V^+} \left\{ p(y \mid x) \right\} \tag{1}$$

which can only be approximately computed in practice using techniques such as greedy search.

In recent years, the increase in performance of LLMs has made sampling also a viable option. For auto-regressive models, this is usually done using ancestral sampling, often with some re-shaping of the token distribution by setting the temperature or nucleus size (Holtzman et al., 2019).

In this context, techniques that aggregate over multiple outputs of the same model have become a simple yet powerful way to further boost results. This is best exemplified by the resurgence of minimum Bayesian risk decoding (MBRD) and related methods applied to LLMs, such as self-consistency (Wang et al., 2023), as well as strong results for "consensus" in reasoning models such as OpenAI's O1, O3 or DeepSeek's R1, which can be interpreted as MBRD with exact match.

MBRD solves an alternative decoding problem, where the goal is to find the output that minimizes the expected risk with respect to the LLM distribution. In the remainder of this manuscript, we will refer instead to the mathematically equivalent problem of maximizing the expected similarity, which makes notation simpler. This problem can be expressed as

$$\hat{y} = \arg \max_{y' \in V^+} \left\{ \mathbb{E}_{p(y|x)} \{ M(y, y', x) \} \right\} = \arg \max_{y' \in V^+} \left\{ Q(y', x) \right\} \tag{2}$$

where $M(y, y', x)$ is a similarity function between outputs $y, y' \in V^+$. [4]. Exact MBRD is doubly intractable since it requires the same search over $V^+$ as greedy decoding, but also the computation

---

[3]For recent results combining step-by-step and post-processing see `https://huggingface.co/spaces/HuggingFaceH4/blogpost-scaling-test-time-compute`

[4]For generality, we have also included the input in this function, since it does not alter the formulation.

of the expectation over that same domain. For this reason, MBRD is often approximated through Monte Carlo estimation by using a set of samples $\mathcal{S}(N) = \{y_1 \cdots y_N\} \sim p(y \mid x)$, both as the search space and to compute the expectation:

$$Q(y', x) = \mathbb{E}_{p(y|x)}\{M(y, y', x)\} \approx \frac{1}{N} \sum_{y_n \in \mathcal{S}(N)} M(y_n, y', x). \tag{3}$$

Often described as *consensus* or *majority voting*, MBRD using exact match similarity, henceforth referred to as MBRD (EM), is a well known and strong baseline

$$M(y, y') = \delta_{g(y,y')}. \tag{4}$$

Here $g()$ is a function that extracts an answer from each model output (which may contain CoT and other tokens), compares them using some normalization, e.g., a symbolic representation, and returns 1 if they are equal or 0 otherwise. This amounts to selecting the answer that occurs more often in this normalized space. Other forms of MBRD include using symbol-level distances such as Rouge (Lin, 2004). In some fields like machine translation, evaluation metrics like BLEU (González-Rubio et al., 2011) or COMET (Guttmann et al., 2024) are also used. For LLMs, it is straightforward to incorporate a reward model into the risk/similarity computation as

$$M(y, y', x) = \tilde{M}(y, y', x) \cdot R(y, x), \tag{5}$$

where $\tilde{M}$ is the original similarity function. This method can also be viewed as an instance of a voting verifier (Li et al., 2023). This setup will henceforth be referred to as MBRD (EM*R).

## 4 Optimal Policy Minimum Bayesian Risk Decoding

### 4.1 Definition

We here propose another way of combining $p(y \mid x)$ and $R(y, x)$ that represents a minimum increase in complexity, while providing interesting properties. Borrowing from Reinforcement Learning, one can define a distribution $q$ that maximizes an expected reward $R(y, x)$ while being close to a reference distribution $p_R(y \mid x)$. This can be expressed as the objective

$$\mathcal{L}(q) = \mathbb{E}_{q(y|x)}\{R(y, x)\} - \beta \cdot \mathrm{KL}(q(y \mid x) \,\|\, p_R(y \mid x)) \tag{6}$$

where $\beta$ controls how much influence the reward has on $q$. This objective is the well known KL-controlled reward maximization, which is the basis for RL algorithms such as PPO (Schulman et al., 2017), GDC++ (Korbak et al., 2022), DPO (Rafailov et al., 2023) and BRAIn (Pandey et al., 2024). It is easy to see that the solution to this is given by the optimal policy [5]

$$p^*(y \mid x) = \arg\max_q \{\mathcal{L}(q)\} = \frac{1}{Z} \cdot p_R(y \mid x) \cdot \exp\left(\frac{1}{\beta} R(y, x)\right) \tag{7}$$

where the partition function $Z$ requires an intractable sum over the space of sentences $V^+$. Assuming that we could sample from this distribution, it's trivial to do MBRD with this optimal posterior

$$\hat{y} = \arg\max_{y' \in V^+} \{\mathbb{E}_{p^*(y|x)}\{M(y, y', x)\}\} \tag{8}$$

This formulation provides a well defined way of integrating a reward $R(y, x)$, a similarity function $M(y, y', x)$, a reference model $p_R(y \mid x)$, and an available generator $p(y \mid x)$.

#### 4.1.1 Computing Expectations with respect to the Optimal Policy

To approximate MBRD expectations we need to sample from an intractable energy model, in particular from the optimal policy. This has been addressed before in the literature but for the purpose of Reinforcement Learning training (DPG, GDC++, RSO (Liu et al., 2024), BRAIn). It can be shown

---

[5]For a formulation, see for example Rafailov et al. (2023) Appendix A.1.

that, given a sample from a proposal distribution, in this case assumed to be our generator $p(y \mid x)$, the probability of the sample $y_n \in \mathcal{S}(N)$ belonging to $p^*(y \mid x)$ is given by[6]

$$p(\text{accept } y_n) = \exp\left(\tilde{R}(y_n, x) - \max_{y'} \tilde{R}(y', x)\right) \qquad (9)$$

where

$$\tilde{R}(y, x) = \frac{R(y, x)}{\beta} + \log \frac{p_R(y \mid x)}{p(y \mid x)} \qquad (10)$$

It seems intuitive that just using the accepted samples to compute the expectation is the best option. However, it is well-known that the Rao-Blackwellized version (Casella & Robert, 1996)[7] of this estimator can use all samples to provide a lower variance estimate. This can further be approximated via importance sampling to yield

$$\hat{Q}(y', x)^{\text{OP}} = \frac{1}{N} \sum_{y_n \in \mathcal{S}(N)} M(y_n, y', x) \cdot \frac{p(\text{accept } y_n)}{\sum_{y'_n \in \mathcal{S}(N)} p(\text{accept } y'_n)} \qquad (11)$$

Since softmax is invariant to shifting the logits by a constant, Rao-Blackwellized rejection sampling in Eq. 11 coincides with self-normalized importance sampling (SNIS) (Bengio & Senécal, 2008) with unnormalized weights $p_n$. We term this last estimator Optimal Policy Minimum Bayesian Risk (OP-MBR) and its maximization OP-MBR Decoding (OP-MBRD). A full derivation can be found in Appendix 11.

## 4.2 OP-MBRD WITH A PROCESS REWARD MODEL

The method introduced here provides a well defined way to integrate a reward model $R(y, x)$, a reference model $p_R(y \mid x)$, and a similarity function $M(y_n, y', x)$ into a decoding strategy for a generator $p(y \mid x)$. It does not prescribe which values should these take. In the case of inference scaling, PRMs estimate the odds that a given partial reasoning leads to the correct answer. For these, the acceptance probability can thus be defined as the product of acceptance of every step, leading to

$$\prod_{t=1}^{T} p(\text{accept } y_{<t+1}^n) = \exp\left(\sum_{t=1}^{T} \frac{\text{PRM}(y_{<t+1}^n, x)}{\beta} + \log \frac{p_R(y_n \mid x)}{p(y_n \mid x)} - M\right) \qquad (12)$$

where $M$ is the sum of maximum $\tilde{R}$ for each step. In practice we normalize the sum of PRM scores by the number of steps $T$. Note that this does not require step-by-step decoding. The outputs are fed to the PRM at the end of generation with appropriate markers i.e. double end of line, and the PRM returns scores for what it considers steps. Another possible interpretation of this formula is ancestral importance sampling of the optimal policy.

## 4.3 EFFICIENT OP-MBRD

Since OP-MBRD Rao-Blackwellized rejection sampling and importance sampling estimators coincide, it may seem that the rejection sampling formulation is redundant. Nevertheless, one can still derive a useful metric from it, the number of expected optimal policy samples for a sample set $\mathcal{S}(N)$

$$\hat{N}^{\text{OP}} = \sum_{y_n \in \mathcal{S}(N)} p_n = \sum_{y_n \in \mathcal{S}(N)} \exp\left(\tilde{R}(y_n, x) - \max_{y'} \tilde{R}(y', x)\right). \qquad (13)$$

Under the rejection sampling interpretation of our estimator, this gives us a measure of how successful our sampling round was, with a higher $\hat{N}^{\text{OP}}$ indicating more samples belong to $p^*$. We can use this to derive an efficient version of OP-MBRD, where we fix a desired number of optimal policy samples $N^{\text{OP}}$ and we sample repeatedly until $\hat{N}^{\text{OP}} \geq N^{\text{OP}}$. As it will be shown in the experimental setup, these yields a good prediction of task difficulty for generator-PRM pairs. We will describe those pairs has being *well calibrated* and henceforth refer to this proposed method as OPE-MBRD.

---

[6]See e.g. RSO (Liu et al., 2024) Appendix A.1

[7]See https://andrewcharlesjones.github.io/journal/rao-blackwellization.html

### 4.4 Formal Guarantees

Unlike other methods that combine majority voting and reward models, like MBRD (EM*R), OP-MBRD has a clearly defined asymptotic behavior, trivially following from the properties of self-normalized importance sampling[8]. The proposed OP-MBRD estimator converges to the true MBRD with respect to the optimal policy with probability 1.

$$p\Big( \lim_{N \to \infty} \hat{Q}(y', x)^{\text{OP}} = \mathbb{E}_{p^*(y|x)}\{M(y, y', x)\} \Big) = 1 \tag{14}$$

Furthermore, we can examine in detail Eqs 35,10 to study what sampling from the optimal policy entails. For cases in which $R = 0$ only the log-ratio term remains and OP-MBRD reduces to MBRD from $p_R(y \mid x)$, as approximated by self-normalized importance sampling. If we use our generator as reference $p_R(y \mid x) = p(y \mid x)$, only the reward term $R$ remains. For a PRM this now will represent MBRD with respect to an energy model proportional to the odds of reaching the correct answer. For an oracle PRM this would assign zero weight to any sample in $S(N)$ not reaching the correct answer, which in the limit guarantees that OP-MBRD always chooses the right answer [9].

## 5 Experimental Setup

### 5.1 Models and Datasets

To evaluate the methods proposed, we test small and medium LLMs on math and coding tasks. For reproducibility and completeness, we select recent open source models in the 1-20 billion parameter range. For math, we select Alibaba's Qwen-2.5-math Instruct models (Abdin et al., 2024) sizes 1.5b and 7b as high performing math-specific models. These have a matching process reward model – Qwen-2.5-PRM-7b (Zhang et al., 2025) – that we also use in our experiments. We also select IBM's Granite 3.3[10] models sizes 2b and 8b. These are general models that also exhibit strong math performance compared to, e.g., LLaMa models of the same size (Grattafiori et al., 2024). For the Granite models, we train our own PRM from Granite-3.3 for math. The model was trained with synthetically generated data. The training data consists of step-level correctness annotations, generated using the binary search method of Luo et al. (2024). The input prompts are sampled from MathInstruct (Yue et al., 2023), MetaMathQA (Yu et al., 2023) and NuminaMath (Li et al., 2024) datasets – the responses are sampled from Granite-3.x, Mixtral-8x22B and Phi4-instruct models. After initial training of the PRM with this data, we use the trained PRM to further filter out low-quality step annotations. We discard samples where step-level correctness annotations do not match the PRM's assessment of step quality. We then perform a second iteration of PRM training with this higher-quality filtered data. As the upper tier in size we select again Phi-4-instruct (Abdin et al., 2024) (14b) as an additional generator. Finally, we pair Phi4-instruct also with a Phi4-PRM trained the same way. We do not include long-CoT models since we are focusing here on approaches leveraging BoN and MBRD, which leverage independent samples rather than long contexts.

We evaluate all models and methods on MATH-500 (Hendrycks et al., 2021; Lightman et al., 2024) and HumanEval (Chen et al., 2021). MATH-500 is a collection of 500 math competition problems requiring step-by-step reasoning to solve. HumanEval consists of 164 programming problems, each asking to complete a standalone Python function from its docstring. Unit tests are included for each example for automatic evaluation. We use pass@1 scores to assess performance on both datasets.

### 5.2 Baselines and Methods

As inference scaling baselines we focus on well established single-step algorithms. We consider BoN using the average PRM score across steps, which was observed to be more performant than other aggregations like minimum in these datasets. We use also two variants of MBRD. First, variants not making use of a PRM or any other parametric scoring function. For MATH-500 we use exact match similarity, $M(y, y') = \delta_{g(y,y')}$. This is often also described as majority voting and is here termed MBRD (EM). As text normalizer $g()$ we extract the answer inside the boxed

---

[8]For a derivation see https://www.tuananhle.co.uk/notes/is.html

[9]Assuming a perfect PRM, $\lim_{N \to \infty}$ and model assigning non zero probability to the solution

[10]https://huggingface.co/ibm-granite/granite-3.3-8b-instruct

command. For code, exact match performs very poorly, we use RougeL (Lin, 2004) instead, $M(y, y') = \text{rougeL}(y, y')$, which performed best in initial tests. We refer to this as MBRD (rouge). As parametric MBRD baseline, we used Voting Verifier (Li et al., 2023), which can be expressed as $M(y, y') = \delta_{g(y,y')} \cdot \text{meanPRM}(y, x)$ here termed MBRD (EM*R).

As methods proposed, we introduce Optimal Policy variants of the non-parametric MBRD i.e. OP-MBRD (EM) and OP-MBRD (rouge). These use the Rao-Blackwellized rejection sampling (or equivalently importance sampling) to sample from the optimal posterior (see Section 4). All our experiments use $p_R(y \mid x) = p(y \mid x)$, which in practice nullifies the effect of the log-ratio term of OP-MBRD. Although the log-ratio term can be proved to equate rejection/importance sampling of $p_R$ (see Section 4.4), initial experiments show no advantage when using strong teachers for $p_R$. We provide an analysis in Appendix 9. This also allowed to estimate the maximum reward in Eq. 35 as the maximum PRM value 1.0. In addition to the normal variants, we also used the efficient version proposed in Section 4.3, termed OPE-MBRD, which uses the expected amount of accepted samples to decide when to stop sampling. For this we iteratively sampled outputs one by one until a target budget of $N^{\text{OP}} = \{1, 2, 4, 8, 16, 32, 64, 128, 256\}$ optimal samples was met. These experiments are designed to measure the gains in throughput and not in wall-clock time. For the latter, a schedule would have to be designed that uses the observed probability of success to guess a fixed number of samples to be generated next. We leave this for future work.

## 5.3 HYPERPARAMETERS AND EXPERIMENT REPETITION

For hyperparameter tuning, we construct a development set out of NuminaMath[11]. We include a random subset of 500 question-answer pairs in this set, discarding their CoTs, and making sure they (a) pass simple format check, and (b) are not in MATH-500. We set the KL-term weight $\beta$, representing the relative weight of the generator versus the PRM in this set. A value of $\beta = 0.1$ was found to be robust across many scenarios and was selected for Qwen, Granite and Phi-4 models both for math and code tasks. The only clear exception was Qwen-1.5b/Qwen-7B-PRM. Results on the dev set indicated that, for this pair, the PRM is much stronger than the generator, and a value of $\beta = 0.001$ was selected. For the OPE-MBRD a maximum number of samples was set as a $\times 10$ multiplier of the number of optimal samples selected. For example, if we solicited 2 optimal samples, no more than 20 real samples would be sampled. This was a simple compromise that helped with badly calibrated generator-PRM pairs, that tend to have spikes in the number of samples solicited. We include the full dev details in Appendix 8.

We investigate inference scaling up to $N = 256$ samples per input. To reduce variance of results, we always make use of the pool of 256 samples for all experiments, either for ensembling or experiment repetition. [12]. Note that for the efficient version of OP-MBRD, OPE-MBRD, the number of samples that constitutes an experiment changes, since the algorithm can select a different number of samples for different generations. For this, we consume blocks of samples of variable size until exhausting the sample pool to construct experiment repetitions. No sample is ever shared across experiments. To obtain a final average and standard deviation, we further repeat the experiments above 3 times.

## 5.4 RESULTS ANALYSIS

Figure 1 shows the comparison of the different generator-PRM pairs. The left shows pass@1 performance as a function of the real number of samples generated, averaged over all dataset examples. The right side shows study cases for specific optimal policy budgets of OPE-MBRD, signaled with a cyan star on the left side of the plot. Each marker on the right represents an example in the MATH-500 dataset, sorted from lower to higher difficulty. The difficulty is assessed by using the pass@1 of the normal generator $p(y \mid x)$ and the full 256 samples. On the vertical axis we display the real number of samples $N$ used by OPE-MBRD averaged over experiment repetitions. We color as green instances for which the OPE-MBRD attains higher pass@1 than OP-MBRD, red if lower, black if both match (typically both reach 1.0) and gray if both fail (0.0). We consider generator-PRM pair as well calibrated if the number of samples used increases with problem difficulty and this leads to performance improvements (green dot). Improvements below 0.02 are ignored to avoid noise.

---

[11]https://huggingface.co/datasets/AI-MO/NuminaMath-CoT

[12]For example, for MBRD (EM) with $N = 16$ samples, we can repeat the experiment and average scores over $256/16 = 16$ repetitions

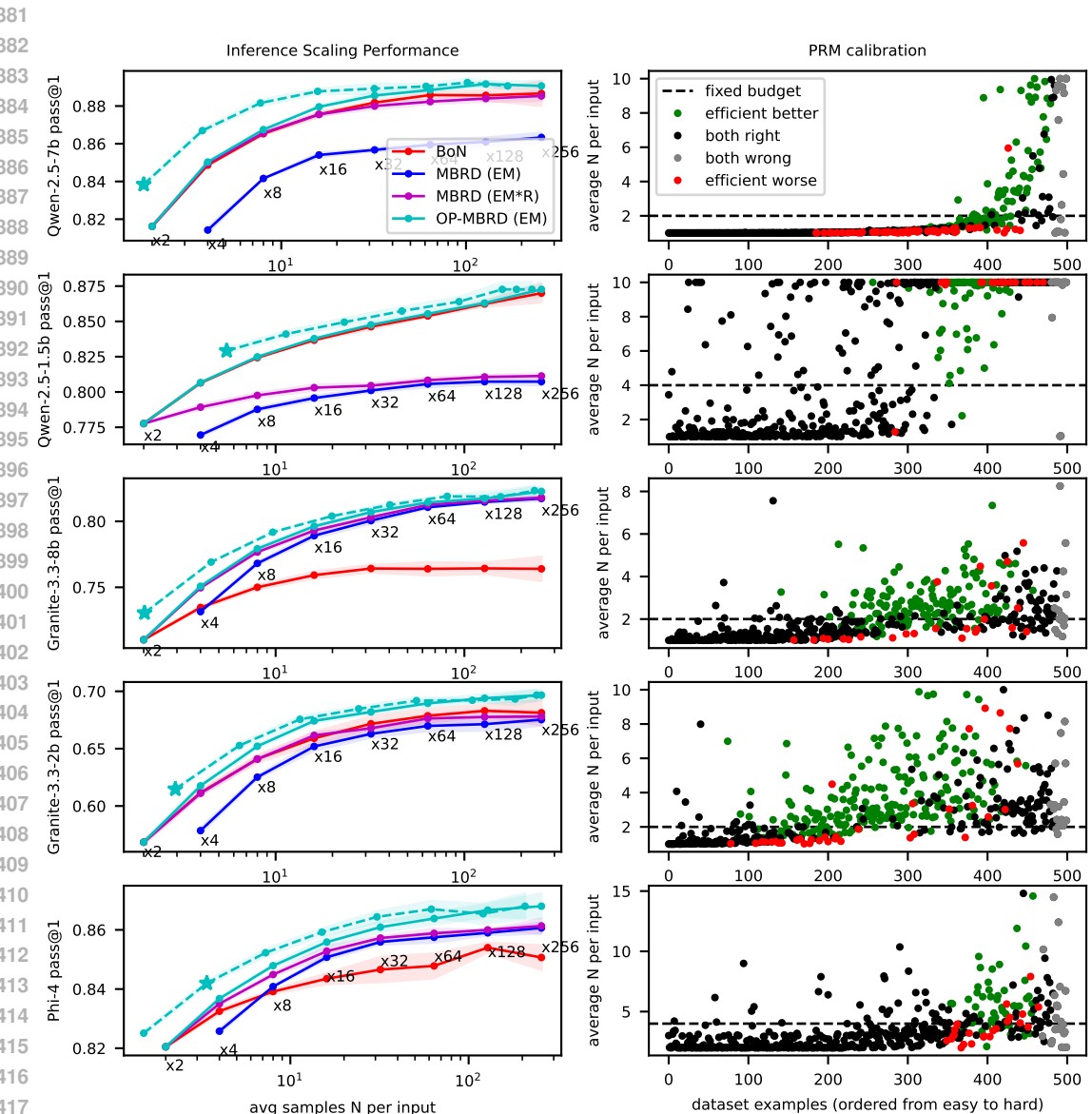

Figure 1: MATH-500 test results. Left: pass@1 score as a function of the number of samples per input with average and standard deviation displayed. A dashed line denotes the efficient version (OPE-MBRD). A star marker denotes the OPE-MBRD configuration represented on the right side. Right: Number of samples OPE-MBRD selects for every example in the test set, sorted from easy to difficult by regular decoding pass@1.

Looking at Figure 1, left: In terms of pass@1 performance, OP-MBRD performs robustly across scenarios and is mostly above or equal to the best baseline, which alternates between BoN or MBRD (EM*R). For the stronger Qwen-7b-math/Qwen-PRM-7B, results match or slightly outperform the baseline MBRD (EM*R). For the efficient version OPE-MBRD, large gains in performance at attained at low numbers of samples- this is consistent with the excellent calibration where the OPE-MBRD version select one real sample for all but the hardest 15% of all examples. In the smaller Qwen case the BoN baseline attains much better performance than MBRD (EM*R) baseline, but

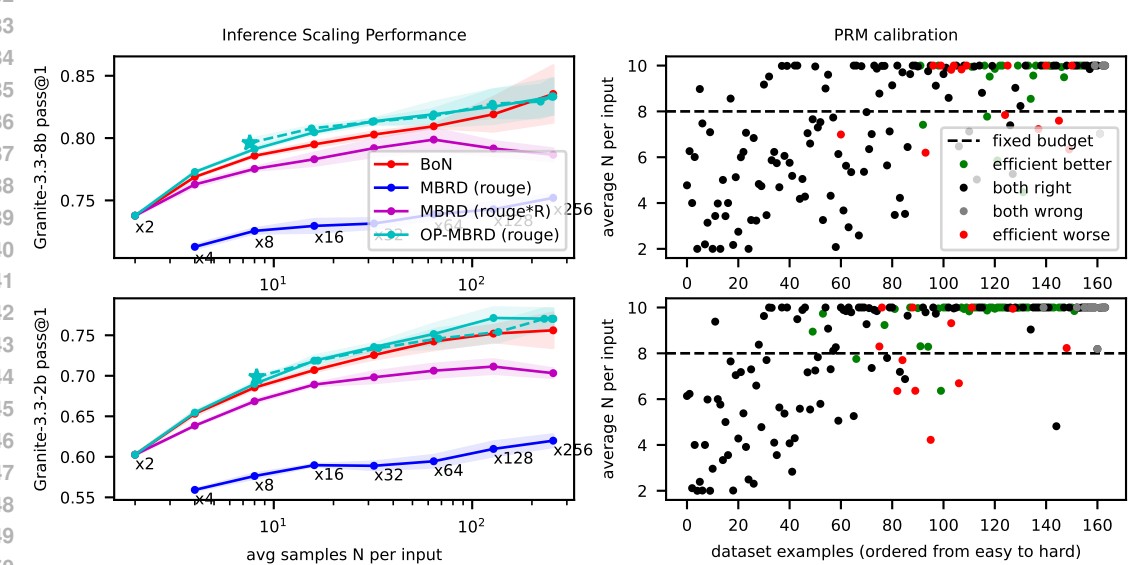

Figure 2: HumanEval test results: Left: pass@1 score as a function of the number of samples per input with average and standard deviation displayed. A dashed line denotes the efficient version (OPE-MBRD). A star marker denotes the OPE-MBRD configuration represented on the right side. Right: Number of samples OPE-MBRD selects for every example in the test set, sorted from easy to difficult by regular decoding pass@1.

OP-MBRD closely matches it. Despite the worse calibration OPE-MBRD still provides a good advantage. For the Granite/Granite-PRM pairs, which are weaker at math, OP-MBRD provides an advantage over the baseline MBRD (EM*R), with particularly strong gains for the smaller model and high number of samples. Both results show reasonably good, but noisier, calibration which leads to OPE-MBRD providing gains over OP-MBRD. Phi-4/Phi-4-PRM presents the worst calibration, which leads to OPE-MBRD just matching OP-MBRD, but overall gaining a small advantage against the best baseline MBRD (EM*R). The lack of Phi-4/Phi-4-PRM calibration may stem from the fact that PRM development was mostly centered around the Granite models.

Figure 2 shows additional results for Granite/Phi-4-PRM pairs[13] on the the HumanEval coding task. All metrics and symbol meanings are same as before. Overall, OP-MBRD remains close to the best performing baseline, in this case BoN. Calibration in this case is non-existent, which can be explained by the fact that we use a PRM tuned on math data to judge a coding task, resulting in very low overall PRM scores and very pessimistic (high) number of samples solicited. As with MATH-500, OP-MBRD still provides an advantage for the smaller model and at higher sample counts.

# 6 CONCLUSIONS

We present Optimal Policy Minimum Bayesian Risk Decoding (OP-MBRD), a simple alternative to BoN and MBRD with rewards that performs more robustly across different generator-PRM combinations. OP-MBRD also has well-defined asymptotic behavior interpolating, in an interpretable way, between rejection/importance sampling from a target generator and sampling from an energy model associated to a reward model. Finally, the proposed formulation also yields an additional useful signal that can suggest a variable number of samples based on input difficulty. For well-calibrated generator-PRM pairs, this results in large gains in throughput for the same compute budget, without relying on answer counts or risk/value functions. Future work can expand on the role of the reference generator and look into efficient multi-step algorithms, for which the properties of the presented method are well-suited.

---

[13]We paired Granite with Phi-4-PRM since it showed better overall performance on the coding task.

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

## 7 LIMITATIONS

The work presented here has the following limitations: Although we cover a diverse set of 3 gener-ators and 3 PRMs covering 5 different sizes across the range of 1-20B parameters, this is not fully representative of all LLMs. In particular, larger models that are likely to have better generation capabilities would be interesting to look at. In this setting, MBRD could be expected to have addi-tional advantages over BoN. Unfortunately, due to compute limitations, it was not possible to cover all such cases. Although we cover both math and coding tasks, we had to keep the scope limited due to both time and compute constraints. In particular, a separate development set for coding and a larger experimental setup would have provided better opportunity to explore the methods better. Other domains where MBRD is also well established, such as machine translation, could also have added value.

## 8 DEVELOPMENT SET RESULTS

As stated in Section 5.3, we created a dev set for hyperparameter tuning based on NuminaMath[14] of the same size as MATH-500. We report full results on the development set under the same conditions as the MATH-500 test set in Figure 3. These results were used to tune $beta$ for different generator/PRM pairs. As it can be observed from the results, this dataset is harder than Math-500, but model/PRM calibration is similar. Overall improvements with OP-MBRD are also larger, but this can be due to tuning effects.

## 9 EFFECT OF REFERENCE POLICY

Figure 4 shows the effect of the log-ratio component on the overall OP-MBRD. In Equation 10, this amounts to setting the reference policy equal to the generator (OP-MBRD), equal to a given reference policy (OP-MBRD + log-ratio) or the latter but setting the reward to 1.0 (log-ratio only). As explained in Section 4.4 the last one should be equivalent to self-normalized importance sampling from the reference model. As reference policy, we use Qwen-2.5-72b-math-Instruct. Different

---

[14]`https://huggingface.co/datasets/AI-MO/NuminaMath-CoT`

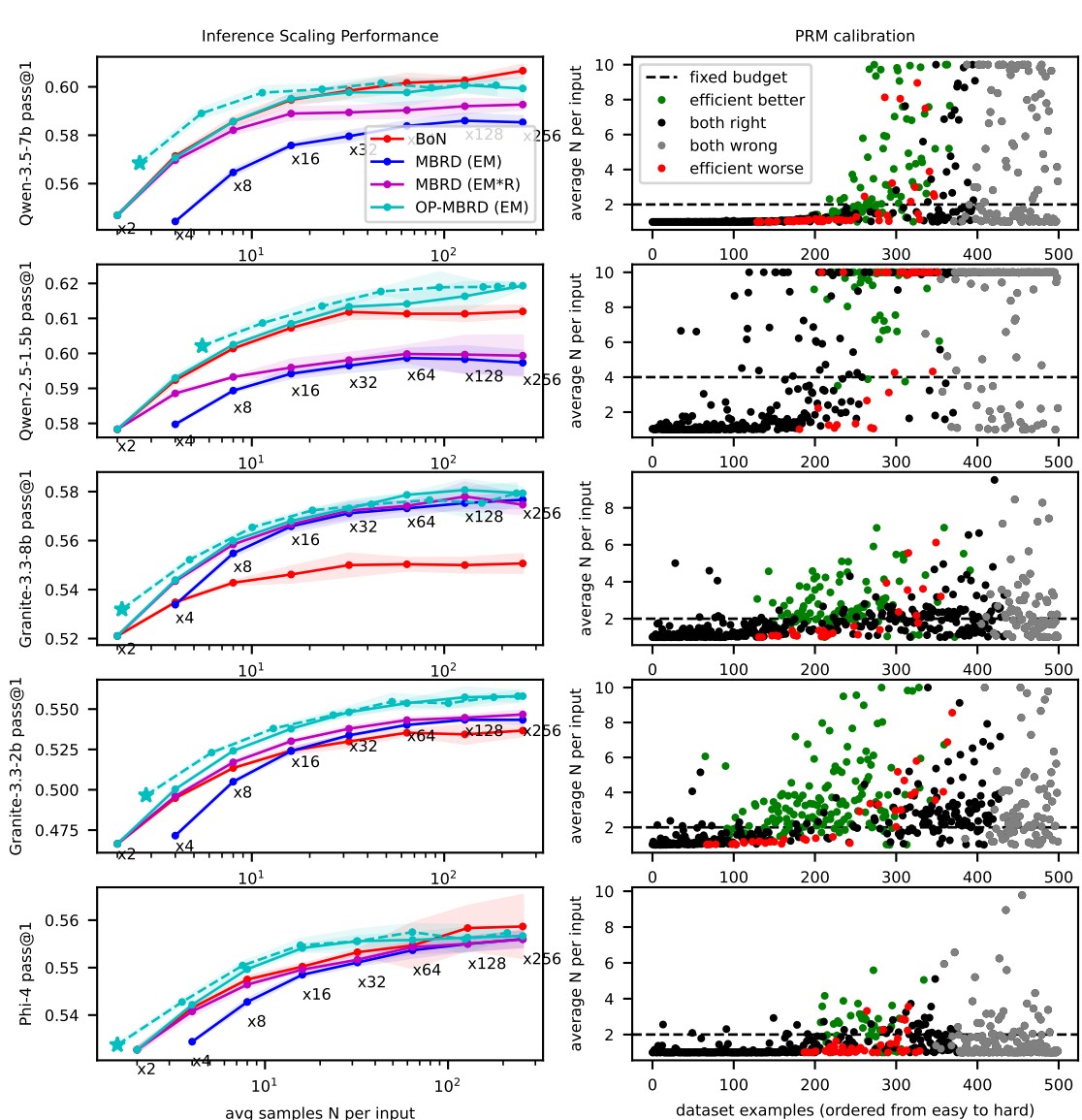

Figure 3: Development set (Numinamath-500 subset) results. Left: pass@1 score as a function of the number of samples per input with average and standard deviation displayed. A dashed line denotes the efficient version (OPE-MBRD). A star marker denotes the OPE-MBRD configuration represented on the right side. Right: Number of samples OPE-MBRD selects for every example in the test set, sorted from easy to difficult by regular decoding pass@1.

values of $\beta$ were tried in the dev set. Log-ratio alone showed no sensitivity to $\beta$ while OP-MBRD + log-ratio resulted in same or degraded performance. We present here results on the math-500 test set. The nominal values for the experimental setup of Section 5 are used here. The log-ratio component alone does show improvements over conventional MBRD without the need for a PRM, but these decrease with model size. When combined with a PRM, the improvement overall is negligible.

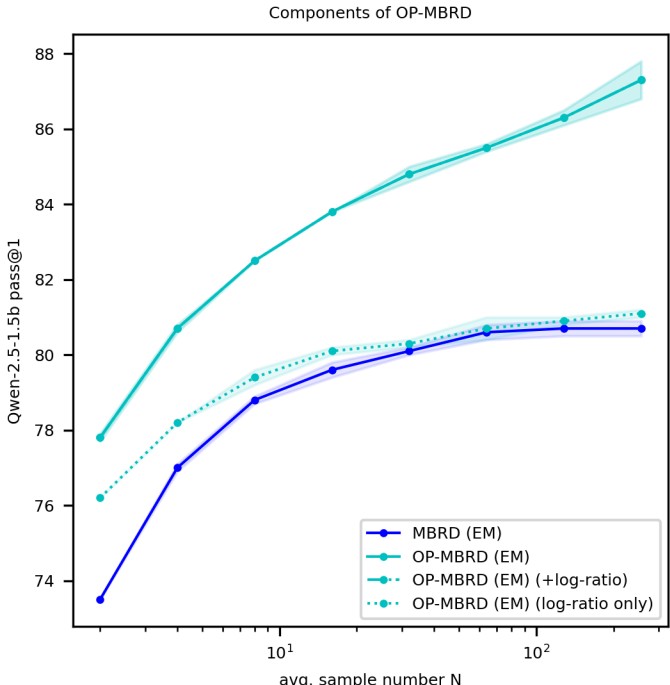

Figure 4: Effect of the reference policy on OP-MBRD. Experimental setup is that of Figure 1 for Qwen-2.5-1.5b-math-Instruct using Qwen-2.5-72b-math-Instruct as reference policy. Note that OP-MBRD (+log-ratio) overlaps with OP-MBRD

## 10 HARDWARE

Runtime experiments were carried out on a private H100 cluster. The code was a fork of math-eval-harness, concretely the one in [15]. Models were served using VLLM[16]. Some steps like computation of MBRD similarity were carried out on standard CPUs. The Phi-4 PRM model training was carried out on 8 H100 GPUs in a private cluster, and inference was done on a single H100 GPU using the Hugging Face `Transformers` library.

## 11 FULL DERIVATION

For ease of reading, and although this is already covered in prior works which are cited in the main body of the paper, we provide here a full derivation of the OP-MBRD algorithm.

### 11.1 OPTIMAL POLICY

The decoding rule is defined as

$$\hat{y} = \arg\max_{y' \in V^+} \{\underbrace{\mathbb{E}_{p^*(y|x)}\{M(y, y', x)\}}_{Q(y', x)}\}$$ (15)

for a $p^*(y \mid x)$ satisfying

$$p^*(y \mid x) = \arg\max_q \{\mathcal{L}(q)\}$$ (16)

---

[15]`https://github.com/QwenLM/Qwen2.5-Math`
[16]`https://github.com/vllm-project/vllm`

with

$$\mathcal{L}(q) = \mathbb{E}_{q(y|x)}\left\{R(y, x)\right\} - \beta \cdot \text{KL}(q(y \mid x) \mid\mid p_R(y \mid x)) \tag{17}$$

where KL is the Kullback-Leibler (KL) divergence and $p_R(y \mid x)$ is a reference policy we want to stay close to. For the purpose of optimization, we can multiply this objective by $1/\beta$ and reformulate it by applying the definition of KL divergence

$$\mathcal{L}(q)/\beta = \mathbb{E}_{q(y|x)}\left\{\frac{R(y, x)}{\beta}\right\} - \text{KL}(q(y \mid x) \mid\mid p_R(y \mid x)) \tag{18}$$

$$= \mathbb{E}_{q(y|x)}\left\{\frac{R(y, x)}{\beta} + \log\frac{p_R(y \mid x)}{q(y \mid x)}\right\} \tag{19}$$

$$= \mathbb{E}_{q(y|x)}\left\{\log\frac{p_R(y \mid x) \cdot \exp\left(\frac{R(y,x)}{\beta}\right)}{q(y \mid x)}\right\} \tag{20}$$

$$= \mathbb{E}_{q(y|x)}\left\{\log\frac{\frac{p_R(y|x) \cdot \exp\left(\frac{R(y,x)}{\beta}\right)}{Z}}{q(y \mid x)}\right\} + \log Z(x) \tag{21}$$

$$= -\text{KL}\left(q(y \mid x) \mid\mid \frac{p_R(y \mid x) \cdot \exp\left(\frac{R(y,x)}{\beta}\right)}{Z}\right) + \log Z(x) \tag{22}$$

$$\tag{23}$$

where have summed and subtracted a partition function to obtain a distance between distributions, which crucially does not depend on $q$

$$\log Z = \sum_{y \in V^+} p_R(y \mid x) \cdot \exp\left(\frac{R(y, x)}{\beta}\right) \tag{24}$$

due to the properties of the KL divergence, $\mathcal{L}(q)$ is maximized $(= 0)$ when the two distributions in the KL match i.e.

$$p^*(y \mid x) = \arg\max_q \left\{\mathcal{L}(q)\right\} = \frac{p_R(y \mid x) \cdot \exp\left(\frac{R(y,x)}{\beta}\right)}{Z} \tag{25}$$

## 11.2 RAO-BLACKWELLIZED REJECTION SAMPLING

For the purposes of solving MBRD, we can approximate the expectation with the following estimator

$$\hat{Q}(y', x) = \frac{1}{N}\sum_{y_n \in S(N)} M(y_n, y', x) \tag{26}$$

where $S(N) = \{y_1 \cdots y_n\} \sim p^*(y \mid x)$. It rests to sample from this distribution to obtain the estimator. For this we use statistical rejection sampling, also known as accept-reject sampling. This uses the fact that if we sample from any proposal $y \sim p(y \mid x)$ the probability of that sample belonging to $p^*(y \mid x)$ is given by

$$p(\text{accept } y_n) = \frac{p^*(y \mid x)}{M \cdot p(y \mid x)} \tag{27}$$

where $M$ is a scaling factor to ensure the ratio is a valid probability. The tightest scaling factor is given by

$$M = \max_y \left\{\frac{p^*(y \mid x)}{p(y \mid x)}\right\} \tag{28}$$

replacing Eq. 25 into Eq 27, this can be reformulated as

$$p(\text{accept } y_n) = \exp\left(\frac{R(y,x)}{\beta} + \log\frac{p_R(y \mid x)}{p(y \mid x)} - \log M\right) \tag{29}$$

with

$$M = \max_y\left\{\exp\left(\frac{R(y,x)}{\beta} + \log\frac{p_R(y \mid x)}{p(y \mid x)}\right)\right\} \tag{30}$$

$$= \exp\left(\max_y\left\{\frac{R(y,x)}{\beta} + \log\frac{p_R(y \mid x)}{p(y \mid x)}\right\}\right) \tag{31}$$

$$\tag{32}$$

renaming as

$$\tilde{R}(y,x) = \frac{R(y,x)}{\beta} + \log\frac{p_R(y \mid x)}{p(y \mid x)} \tag{33}$$

yields

$$p(\text{accept } y_n) = \exp\left(\tilde{R}(y_n,x) - \max_{y'}\tilde{R}(y',x)\right) \tag{34}$$

A naive estimator based on these probabilities can be built using an indicator function and samples from a uniform distribution $u_n \sim U(0,1)$

$$I_n = \delta_{p(\text{accept } y_n) \geq u_n} \tag{35}$$

which will select a set of samples from the proposal $p(y \mid x)$ using the acceptance probabilities. This results in

$$\hat{Q}(y',x)^{\text{RS}} = \frac{1}{N}\sum_{y \in S(N)} I_n \cdot M(y_n,y',x) \tag{36}$$

However, the Rao-Blackwell estimator based on the sufficient statistic $p_1 \cdots p_N$, which involves all samples rather than just the accepted ones, can be proven to be equal or lower variance.

$$\hat{Q}(y',x)^{\text{OP}} = \mathbb{E}\{Q(y',x) \mid p_1 \cdots p_N\} \tag{37}$$

For rejection sampling this can be approximated by self-normalized importance sampling using the relation on Eq 27. For normal importance sampling we have that

$$Q(y',x) = \mathbb{E}_{p^*(y|x)}\{M(y,y',x)\} \tag{38}$$

$$= \sum_{y \in V^+} p^*(y \mid x) \cdot M(y,y',x) \tag{39}$$

$$= \sum_{y \in V^+} p(y \mid x) \cdot M(y,y',x) \cdot \frac{p^*(y \mid x)}{p(y \mid x)} \tag{40}$$

$$= \sum_{y \in V^+} p(y \mid x) \cdot M(y,y',x) \cdot p(\text{accept } y) \cdot M \tag{41}$$

$$\tag{42}$$

with the scaling factor $M$ canceling after self-normalization leading to the OP-MBRD estimator expression

$$\hat{Q}(y', x)^{\text{OP}} = \sum_{y_n \in S(N)} M(y_n, y', x) \cdot \frac{p(\text{accept } y_n)}{\sum_{y_n \in S(N)} p(\text{accept } y_n)} \tag{43}$$

$$= \sum_{y_n \in S(N)} M(y_n, y', x) \cdot \frac{\exp\left(\tilde{R}(y_n, x) - \max_{y''} \tilde{R}(y'', x)\right)}{\sum_{y_m \in S(N)} \exp\left(\tilde{R}(y_m, x) - \max_{y''} \tilde{R}(y'', x)\right)} \tag{44}$$

### 11.3 IMPORTANCE SAMPLING EQUIVALENCE AND EXPECTED NUMBER OF OPTIMAL POLICY SAMPLES

ne can also use importance sampling directly in the place of rejection sampling, for the non normalized version

$$Q(y', x) = \mathbb{E}_{p^*(y|x)}\{M(y, y', x)\} \tag{45}$$

$$= \sum_{y \in V^+} p^*(y \mid x) \cdot M(y, y', x) \tag{46}$$

$$= \sum_{y \in V^+} p(y \mid x) \cdot M(y, y', x) \cdot \frac{p_R(y \mid x) \cdot \exp\left(\frac{R(y, x)}{\beta}\right)}{Z \cdot p(y \mid x)} \tag{47}$$

$$= \sum_{y \in V^+} p(y \mid x) \cdot M(y, y', x) \cdot \frac{1}{Z} \cdot \exp\left(\frac{R(y, x)}{\beta} + \log \frac{p_R(y \mid x)}{p(y \mid x)}\right) \tag{48}$$

$$\tag{49}$$

after self-normalization this leads to

$$\hat{Q}(y', x) = \sum_{y_n \in S(N)} M(y_n, y', x) \cdot \frac{\exp\left(\tilde{R}(y_n, x)\right)}{\sum_{y_m \in S(N)} \exp\left(\tilde{R}(y_m, x)\right)} \tag{50}$$

which only differs from the previous estimator by a constant inside of the exponential (the scaling factor), which cancels out. Both estimator are therefore identical.

One important difference in practice is that we can derive the expected number of samples that are accepted, which is used in OPE-MBRD as the stopping condition.

$$\hat{N}^{\text{OP}} = \sum_{y_n \in \mathcal{S}(N)} p_n = \sum_{y_n \in \mathcal{S}(N)} \exp\left(\tilde{R}(y_n, x) - \max_{y'} \tilde{R}(y', x)\right). \tag{51}$$

### 11.4 PRM VERSION

The estimator presented above could be used both with Outcome Reward Models (ORM), and Process Reward Models (PRM). We focused on the latter for our experimental set up and we also set $p_R(y \mid x) = p(y \mid x)$. PRMs compute one score per step of thought and thus we need to compute the probabilities for each

$$p(\text{accept } y_{<t+1}^n) = \exp\left(\frac{\text{PRM}(y_{<t+1}^n, x)}{\beta} - \max_{\tilde{y}_{<t+1}^n} \frac{\text{PRM}(\tilde{y}_{<t+1}^n, x)}{\beta}\right) \tag{52}$$

This leads to a special case detailed here.

$$\tilde{R}(y, x) = \frac{\text{PRM}(y_{<t+1}^n, x)}{\beta} + \underbrace{\log \frac{p(y \mid x)}{p(y \mid x)}}_{=0} \tag{53}$$

since PRMs are bounded by $1.0$ we used this value as the maximum and normalized by the length, leading to

$$p(\text{accept } y^n) = \prod_{t=1}^{T} p(\text{accept } y_{<t+1}^n)^{1/T} = \exp\left(\frac{1}{\beta} \cdot \frac{1}{T} \cdot \sum_{t=1}^{T} (\text{PRM}(y_{<t+1}^n, x) - 1)\right) \tag{54}$$

