# OpenReview forum: "Optimal Policy Minimum Bayesian Risk"
_ICLR.cc/2026/Conference — Submitted to ICLR 2026_

### Official Review · Reviewer_QbyP · 2025-10-28

**Soundness:** 3
**Presentation:** 3
**Contribution:** 3
**Rating:** 6
**Confidence:** 4

**Summary:**

The paper introduces Optimal Policy Minimum Bayesian Risk Decoding , a new inference-time decoding algorithm for large language models. Building on Minimum Bayes Risk Decoding (MBRD), OP-MBRD integrate reward models and similarity measures through a KL-controlled reinforcement learning framework. This method aim to enhance robustness, accuracy, and efficiency in reasoning and coding tasks. Experiments on MATH-500 and HumanEval datasets demonstrate that OP-MBRD often matches or outperforms existing methods like Best-of-N sampling and MBRD with rewards while maintaining computational efficiency and interpretability.

**Strengths:**

- The integration of MBRD with a KL-regularized optimal policy formulation is elegant and well-grounded in reinforcement learning theory, providing clear asymptotic guarantees.
- OP-MBRD consistently achieve strong or superior performance across various LLM–PRM pairs, particularly with sample-efficient variants that adapt sampling to task difficulty.
- The method remains simple to implement, requiring minimal additional parameters, and is compatible with existing reward models, making it widely applicable to different LLMs and domains.

**Weaknesses:**

- Overall, this work is good in theoretical parts. Yet, the study primarily tests small to mid-sized open-source models and focuses on math/coding tasks. This limits the generalizability to larger or multimodal ones like benchmark on Geometry3k.
- The effectiveness of OPE-MBRD depends heavily on the calibration quality between the generator and the reward model. Poorly calibrated pairs, such as Phi-4 with Phi-4-PRM, yield weaker gains.

**Questions:**

see weakness

---

> ### Author Response · Authors · 2025-11-14
>
> We thank the reviewer for their work.
>
> > Yet, the study primarily tests small to mid-sized open-source models and focuses on math/coding tasks. This limits the generalizability to larger or multimodal ones like benchmark on Geometry3k.
>
> This is a valid criticism. One argument in defense of our choice of domains is that inference scaling has mostly shown advantage in math and coding tasks which makes the experimental setup a good match still.
>
> Given the large number of samples needed to perform the experiments we were limited in the models that could be used. In general MBRD/Majority Voting and BoN are known to work at bigger sample sizes. For bigger models it is reasonable to expect high generation quality and therefore a self-consistency signal (i.e. MBRD) to become stronger than the BoN. OP-MBRD is shown to be able to handle both scenarios where one or the other is stronger, so one can hypothesize the results will hold.
>
> > Poorly calibrated pairs, such as Phi-4 with Phi-4-PRM, yield weaker gains.
>
> This is true only for code, which is not the domain where the PRM was trained. Training a PRM for code would likely improve the calibration. On the other hand, well calibrated Generator/PRM pairs such as Qwen provide really good gains and prediction of task difficulty.

---

### Official Review · Reviewer_ys5d · 2025-10-29

**Soundness:** 2
**Presentation:** 1
**Contribution:** 1
**Rating:** 2
**Confidence:** 3

**Summary:**

This paper proposes a new test-time decoding strategy based on minimum Bayesian risk decoding (MBRD) and evaluates it on MATH500 and HumanEval.

**Strengths:**

N/A

**Weaknesses:**

The contributions and empirical results are underwhelming. Conceptually, the proposed OP-MBRD applies the weight $\frac{p_R}{p}\exp(R/\beta)$ (close to the form of the KL-regularized optimal policy) multiplied by a comparison function M. In the MATH500 experiments, M is a delta function and $p_R=p$, and Eq. 9–11 seems to imply that the strategy is simply maximizing $R(x,y)/\beta + \log(\text{empirical frequency of the answer part of} y)$. It is unclear how this meaningfully differs from BoN. It is also possible that Eq. 9-11 are not presented properly (the presentation of Section 4 is not very clear in my opinion).

The experimental comparison among BoN, MBRD-EM (majority voting), MBRD-EM*R (majority voting with PRM), and OP-MBRD-EM (proposed) is weak, and it is only performed on MATH500 and HumanEval. There is no analysis of why OP-MBRD-EM can outperform BoN; any gains may reflect the adaptivity of OP-MBRD or errors in the reward model R.

**Questions:**

See above.

---

> ### Author Response · Authors · 2025-11-14
>
> We would like to thank the reviewer for their work.
>
> > It is unclear how this meaningfully differs from BoN.
>
> The main difference with respect to BoN is that the proposed method performs Minimum Bayesian Risk Decoding (MBRD), of which Majority Voting is a specific instance (i.e. when we use equality as similarity distance).
>
> MBRD is fundamentally different from BoN since it uses the consistency between model answers and not any external reward signal. In the proposed OP-MBRD we use the reward (and a reference model) to modify the generation distribution. This results on a method that uses both consistency between model answers and a reward (unlike BoN, which only uses the reward). This method is most similar to Voting Verifiers / Weighted Majority Voting which it outperforms or matches in all scenarios.
>
> We will improve the main algorithm description to make it more understandable. If the paper were to be accepted, we will have one extra page to add more detailed explanations. We had a full self-contained derivation of the algorithm in `Appendix 11`  to which we will also add citations for specific methods such as rejection and importance sampling. We think this will improve readability notably.
>
> >There is no analysis of why OP-MBRD-EM can outperform BoN;
>
> As per the above explanation. A clear situation in which one can expect OP-MBRD to outperform BoN is when the consistency between answers has is a strong signal while the reward score is not. This may depend from case to case but OP-MBRD shows that it can harness both signals to provide overall greater robustness.

---

> ### Comment · Reviewer_ys5d · 2025-11-25
>
> Thank you for the response. It appears to confirm my earlier point: the proposed method effectively maximizes $R(x,y)/\beta + \text{consistency of } y$, which in my view is a simple twist on BoN with marjority voting. It is plausible that this modification of BoN can improve the performance, but the experiments are far from being comprehensive to establish this.
>
> Currently, the paper focuses on the proposed generic framework, but the presentation is confusing and disorganized. As a result, the work has neither a clear conceptual contribution nor a convincing empirical study. I therefore maintain my evaluation that it is below the ICLR bar.

---

### Official Review · Reviewer_cnDM · 2025-11-05

**Soundness:** 3
**Presentation:** 3
**Contribution:** 2
**Rating:** 4
**Confidence:** 4

**Summary:**

The authors propose a method called Optimal Policy Minimum Bayesian Risk Decoding (OP-MBRD), which builds upon minimum Bayes risk decoding (MBRD) to enhance the inference scaling of large language models (LLMs). Unlike MBRD, which relies on majority vote counting, OP-MBRD incorporates reward and risk/similarity signals into the MBRD framework. It retains much of the simplicity of both the Best of N (BoN) and MBRD methods while introducing only a single new parameter and remaining compatible with general MBRD. Furthermore, the authors demonstrate the efficiency of their Rao-Blackwellized rejection sampling method and provide formal guarantees for it. Empirically, they compare their proposed algorithm with two baselines: BoN and MBRD, using the Math500 dataset and show that their proposed algorithm performs better.

**Strengths:**

- The problem the authors are attempting to address is very important.
- The idea of improving MBRD with Rao-Blackwellized rejection sampling is very interesting.
- The paper is well-written and easy to follow.

**Weaknesses:**

- Although incorporating the closed form of the optimal policy into MBRD is novel, there has been other research that has utilized the tilted distribution as a way to improve policy performance during test time. The authors did not compare their work to these existing studies.
- The paper does not cite relevant work regarding the derivations in certain sections. For example, when discussing Maximum Entropy (MaxEnt), it would be appropriate to cite Ziebart, B. D. for "Maximum Entropy Inverse Reinforcement Learning" or Grünwald, P. D. for "Maximum Entropy, Minimum Discrepancy, and Robust Bayesian Decision Theory."
- Basic naive baselines, such as simple importance sampling, are missing as comparisons to the proposed method when attempting to tilt the distribution.
- As the average number of samples per input increases, the difference between baselines, such as BoN and the proposed methods, decreases. However, we would expect the opposite effect.

**Questions:**

- How does basic importance sampling using the closed form of the optimal compare with the proposed method empirically? Essentially, performing importance sampling as \(\pi_t \exp(Q) / \pi_t = \exp(Q)\) allows for the application of Self-Normalized Importance Sampling (SNIS) or Clipped Importance Sampling (IS).
- The experiments do not compare against a weight-majority vote, which is a strong baseline for inference time scaling.
- The empirical results show pass@1, but pass@k—where k is greater than 1—is extremely important to demonstrate in order to understand the performance of the policy. The plots in the paper may be showing this, but it is unclear what 'pass@1' on the y-axis means when the average samples per input increase.
- The paper is missing important citations, such as:
  - "Controlled Decoding from Language Models" by Mudgal et al.
  - "Value-Guided Search for Efficient Chain-of-Thought Reasoning" by Wang et al.
  - "Value-Augmented Sampling for Language Model Alignment and Personalization" by Han et al.
  - "Inference-Time Language Model Alignment via Integrated Value Guidance" by Qiao et al.

---

> ### Author Response · Authors · 2025-11-14
>
> We would like to thank the reviewer for their work.
>
> >Although incorporating the closed form of the optimal policy into MBRD is novel, there has been other research that has utilized the tilted distribution as a way to improve policy performance during test time. The authors did not compare their work to these existing studies.
>
> We thank the reviewer for bringing our attention to the missed references. We decided to cite FUDGE, a precursor that influences these. In the final paper, we will include all the provided citations and contextualize out claims better given the additional references (see below).
>
> The main argument for not including the referred methods in the experimental setup is that these methods include steps that add complexity to the process (mentioned in S2).  OP-MBRD uses a conventional generator and a single PRM post-processing step (i.e, the scorer is only called once). This is far simpler than most of those methods and has important practical implications for latency.
>
> The FUDGE-inspired approaches referred ("Controlled Decoding", "Value-Guided Search", "Guide Speculative Inference", "Inference-Time Language Model Alignment") as well as the re-sampling approach ("Reward-Augmented Decoding") all operate step by step. This adds following complexities:
>
> 1. separate scorers (extrinsic rewards), are called every decoding step, which incurs a communication overhead between models and client e.g. beam
> 2. customized decoding algorithms to allow token by token, or chunk-level, distribution modifications
> 3. value heads (intrinsic rewards) are model dependent and require RL training e.g. Controlled Decoding
>
> due to these limitations, these kind of methods are not that often used in real scenarios. Given this we considered that well established approaches like MBRD (Majority Voting, Voting Verifiers / Weighted Majority voting) and BoN are the fair baselines.
>
> Regarding a better framing of the contribution, we will refer all the works that the reviewer has indicated, in the context of which we see the main contributions of our work to be:
>
> 1. Aforementioned connection of the optimal policy with MBRD
> 2. Simplicity of the method with a single generation and scoring step
> 3. We also show that the reference/SNIS component helps but is not needed in this setting (see below)
> 4. Consistently better or equal than the best performing method out of BoN or Weighted Majority Voting (Voting Classifier)
> 5. Derivation of the Rao-Blackwellized estimator and resulting efficient version, which shows the ability to predict question difficulty Fig 1, right side.
>
> > Basic naive baselines, such as simple importance sampling, are missing
>
> We do have this result but unfortunately they were left in the `Appendix 9 Fig. 4 (log-ratio only is SNIS)` due to space limitations. These results can now be added with the extra page space if the paper is accepted. While SNIS does improve over MBRD, it is very far from the OP-MBRD improvements. We have two hypothesis
>
> 1. Limited diversity/performance of the proposal. Even with a large number samples, the proposal may not have enough diversity of produce good quality samples for the reference to up scorer, leading to the seen saturation
>
> 2. SNIS bias may also be hurting overall.
>
> >As the average number of samples per input increases, the difference between baselines, such as BoN and the proposed methods, decreases. However, we would expect the opposite effect.
>
> We are not sure what is the hypothesis is for this assumption, but assuming its is a bigger role of SNIS component, as per above this would have a negligible effect. As shown in `Appendix 9 Fig. 4` the PRM is the dominating factor in our setups.
>
> > Q1: How does basic importance sampling using the closed form of the optimal compare with the proposed method empirically?
>
> See response above. It performs clearly worse than OP-MBRD.
>
> > Q2: The experiments do not compare against a weight-majority vote, which is a strong baseline for inference time scaling.
>
> to the best of our knowledge this is the same a Voting Verifiers here termed MBRD(R). This method is outperformed by OP-MBRD.
>
> > Q3: The empirical results show pass@1, but pass@k—
>
> This is a good point, please see next official comment for pass@k results. We will add these to the Appendix
>
> > it is unclear what 'pass@1' on the y-axis means when the average samples per input increase
>
> Assume we have a fixed pool of 256 samples when "samples per input" is N=32, it means all methods use 32 samples for the MBRD/BoN scoring/voting/selection leading to one single experiment outcome for which we compute the exact match. This experiment is then repeated 256/32 = 8 times using the rest of the samples in the pool (no sample re-use). Averaging exact match over 8 repetitions yields the pass@1. See `S5.3` for more details.
>
> > Q4: The paper is missing important citations, such as:
>
> we will include all and contextualize claims better, see above.
>
> Please let us know if you have further questions.

---

> > ### Author Response · Authors · 2025-11-14
> >
> > Related to questions above: pass@k for a reduced number of models and samples on math500
> >
> > Qwen-2.7-7b pass@k
> >
> > `k=1`
> >
> > | N  |   MBRD   |   BoN    | MBRD(EM*R) | OP-MBRD  |
> > |:--:|:--------:|:--------:|:----------:|:--------:|
> > | 2  | 75.1±0.1 | 81.6±0.1 |  81.6±0.1  | 81.6±0.1 |
> > | 4  | 81.4±0.2 | 84.9±0.0 |  85.0±0.1  | 85.0±0.0 |
> > | 8  | 84.2±0.0 | 86.5±0.1 |  86.6±0.1  | 86.7±0.1 |
> > | 16 | 85.4±0.1 | 87.6±0.1 |  87.6±0.1  | 88.0±0.1 |
> >
> > `k=4`
> >
> > | N  |   MBRD   |   BoN    | MBRD(EM*R) | OP-MBRD  |
> > |:--:|:--------:|:--------:|:----------:|:--------:|
> > | 2  | 87.1±0.2 | 89.3±0.0 |  89.3±0.0  | 89.3±0.0 |
> > | 4  | 88.1±0.0 | 90.7±0.1 |  90.5±0.1  | 90.5±0.1 |
> > | 8  | 88.4±0.1 | 91.4±0.1 |  90.8±0.1  | 91.0±0.1 |
> > | 16 | 88.4±0.2 | 91.8±0.1 |  90.9±0.1  | 91.4±0.1 |
> >
> > `k=8`
> >
> > | N  |   MBRD   |   BoN    | MBRD(EM*R) | OP-MBRD  |
> > |:--:|:--------:|:--------:|:----------:|:--------:|
> > | 2  | 90.1±0.1 | 91.5±0.1 |  91.5±0.1  | 91.5±0.1 |
> > | 4  | 90.2±0.0 | 92.5±0.1 |  92.3±0.1  | 92.4±0.1 |
> > | 8  | 90.0±0.0 | 93.1±0.1 |  92.4±0.1  | 92.7±0.0 |
> > | 16 | 89.8±0.3 | 93.1±0.3 |  92.1±0.3  | 92.7±0.2 |
> >
> > Phi-4 pass@k
> >
> > `k=1`
> >
> > | N  |   MBRD   |   BoN    | MBRD(EM*R) | OP-MBRD  |
> > |:--:|:--------:|:--------:|:----------:|:--------:|
> > | 2  | 79.9±0.0 | 82.0±0.0 |  82.0±0.0  | 82.0±0.0 |
> > | 4  | 82.6±0.1 | 83.3±0.1 |  83.5±0.1  | 83.7±0.0 |
> > | 8  | 84.1±0.0 | 83.9±0.1 |  84.5±0.1  | 84.8±0.1 |
> > | 16 | 85.1±0.1 | 84.4±0.2 |  85.3±0.2  | 85.6±0.2 |
> >
> > `k=4`
> >
> > | N  |   MBRD   |   BoN    | MBRD(EM*R) | OP-MBRD  |
> > |:--:|:--------:|:--------:|:----------:|:--------:|
> > | 2  | 88.4±0.1 | 88.9±0.0 |  88.9±0.0  | 88.9±0.0 |
> > | 4  | 88.4±0.2 | 89.1±0.1 |  88.8±0.1  | 88.8±0.1 |
> > | 8  | 88.2±0.0 | 89.0±0.2 |  88.5±0.1  | 88.7±0.2 |
> > | 16 | 88.1±0.2 | 89.0±0.2 |  88.3±0.3  | 88.3±0.4 |
> >
> > `k=8`
> >
> > | N  |   MBRD   |   BoN    | MBRD(EM*R) | OP-MBRD  |
> > |:--:|:--------:|:--------:|:----------:|:--------:|
> > | 2  | 91.0±0.1 | 91.0±0.0 |  91.0±0.0  | 91.0±0.0 |
> > | 4  | 90.4±0.3 | 91.0±0.2 |  90.6±0.2  | 90.5±0.1 |
> > | 8  | 89.6±0.1 | 90.7±0.2 |  89.8±0.2  | 89.8±0.2 |
> > | 16 | 89.0±0.1 | 90.7±0.5 |  89.1±0.3  | 89.1±0.4 |

---

### Official Review · Reviewer_Tk67 · 2025-11-07

**Soundness:** 3
**Presentation:** 2
**Contribution:** 2
**Rating:** 4
**Confidence:** 3

**Summary:**

This paper proposes a method to integrate reward models into the sampling process of an LLM. The authors assume a reference function as well as a reward function, and use a commonly used KL-regularized objective to arrive at an algorithm for picking the best response from the LLM. The proposed method uses a self-normalized importance sampling to find the best candidate. The authors show favorable properties of this method such as consistency. Furthermore, they show competitive experimental performance with voting verifiers.

**Strengths:**

* The background section up to page 4 is well written.
* The method has theoretical derivation.
* The experimental results show competitive performance with voting verifiers.

**Weaknesses:**

* First, the reference model is discussed in all the sections, but it is set to the original model in the experiments. This makes most of the fractions in the reward model equal to 0, and simplifies most of the equations. So, the existence of a reference model is not really justified in the derivations. Also, in practice, it is not really common-practice to set it to a stronger teacher model, so in practice we also do not have a different reference model.
* The paper is only not self-contained, and references to many papers for derivations. For instance, it is not clear from the cited reference how Eq (9) is derived, given the notation difference, and extra factors (e.g,. should y' in the summation have constraints?)
* The writing in section 4.3 and its subsequent experimental results are not easy to follow, this section needs more elaboration.
* The method depends on a hyperparameter $\beta$, and rarely outperforms the voting verifiers (which does not require any hyperparameter).

**Questions:**

* The paper seems to be generalizable to any reward model, why is process reward model chosen?
* In line 248, is the maximum over each step, or over each "sequence"?)
* I would suggest the authors give explicit examples of what x, y, y' is in a given context (such as math examples and final answers), so it would be more accessible to the general audience.
* How sensitive is the overall algorithm to the hyperparameter $\beta$?

---

> ### Author Response · Authors · 2025-11-14
>
> We would like to thank the reviewer for his work. We agree on most issues raised and think missing details/experiments that we provided in the appendix due to space limitations can help clarify them. If the paper were to be accepted, the extra page, together with reviewer feedback, should allow to improve the paper substantially.
>
> >So, the existence of a reference model is not really justified in the derivations
>
> The derivations show that the reference model term makes OP-MBRD asymptotically equivalent to doing MBRD with the reference model w/o needing to sample from it (via the Self-Normalized Importance Sampling (SNIS), if we ignore the PRM, see`S4.4`). In other words, it provides justification for why sampling from a cheap proposal and using an expensive reference could be advantageous. Empirically we show in the `Appendix 9 Fig. 4 (log-ratio only is SNIS)` that the reference model has indeed an effect, but is negligible once the PRM is incorporated. The asymptotic convergence to reference model MBRD via SNIS is a theoretical result that we think is interesting, even if empirically it has not shown an advantage in our setup. Reasons for the limited effect of SNIS in our setup could be limited diversity/performance of the proposal, which explains the observed saturating behavior with increasing N and negative effects of the self-normalization bias. With the extra page we should be able to move this to the main body. We could also add additional experiments that try to extract more from SNIS alone.
>
> > Also, in practice, it is not really common-practice to set it to a stronger teacher model, so in practice we also do not have a different reference model.
>
> we do agree that the regularization role played by the reference model makes much more sense in RL-training than in generation. The SNIS relation does however provide a justification for its use still.
>
> > The paper is only not self-contained, and references to many papers for derivations.
> > The writing in section 4.3 and its subsequent experimental results are not easy to follow, this section needs more elaboration.
>
> This aspect should definitely be improved. We had added the full self-contained derivation of the algorithm `Appendix 11` but we understand also citing scholarly sources for rejection/importance sampling concepts is needed. If the paper would be accepted, it should be easy to expand current 4.1+ explanations with more parts of `Appendix 11` to make it self-contained and also cite references. We can also use extra space to improve the explanations on 4.3.
>
> >The method depends on a hyperparameter $\beta$, and rarely outperforms the voting verifiers (which does not require any hyperparameter).
>
> The sensitivity to the $\beta$ hyperparamenter is low. As reported, all results with the exception of `Fig1. Qwen-2.5-1.5b (math)`) use the same $0.1$. This was set based on the  math dev set but generalizes to both math and code test sets. In retrospect, we should have highlighted better the main advantage of OP-MBRD: It is more robust that either MBRD, MBRD(R) henceforth named Voting Classifier (VC) or BoN. See for example
>
> - BoN outperforms VC/MBRD (by a large margin) for `Fig. 1 Qwen-2.5-1.5b (math)`, while OP-MBRD still outperforms it
> - BoN outperforms VC/MBRD  for `Fig. 2 Granite-3.3-8b (code)` for both small and large models sizes and OP-MBRD still outperforms it
> - VC outperforms BoN/MBRD for `Fig. 1 Granite-3.3-8b (math)` or Phi-4 and OP-MBRD still outperforms it
>
> In total BoN wins over VC **3/7** times, VC wins over BoN **2/7** and the remainder **2/7** they seem matched. In all cases OP-MBRD is above the best of the two or matches the best.
>
> > Q1: The paper seems to be generalizable to any reward model, why is process reward model chosen?
>
> The methods presented should indeed work for any RM with no further changes. PRMs are bounded by 1.0 so it makes implementation and tuning of $\beta$ on dev easier. They also seem to be the usual choice in inference scaling.
>
> > Q2: In line 248, is the maximum over each step, or over each "sequence"?)
>
> Yes and as it is customary in Rejection Sampling (e.g. RSO) we use the sample estimate for this. This was explained above `Eq 3` but we should state it there as well.
>
> > I would suggest the authors give explicit examples of what x, y, y' is in a given context (such as math examples and final answers), so it would be more accessible to the general audience.
>
> we will follow this suggestion thank you.
>
> >How sensitive is the overall algorithm to the hyperparameter ?
>
> It is low, see comment about beta above. We will clarify this better.
>
> Please let us know if anything is not clear or you want further information.

---

> > ### Comment · Reviewer_Tk67 · 2025-11-26
> >
> > Thank you for your response. Overall, I share similar concerns as reviewer ys5d. At its current state, the paper's audience is not very clear. Currently, I can see the paper going three directions: (1) If the authors' target audience is the practical reward modeling community, they should be very clear about how their algorithm works (by adding a figure and an algorithm box, for instance) and show comprehensive results on different datasets. (2) If the authors' target audience is a theoretically sound response selection method, the results should not have too many theoretical jumps and be more self-contained. (3) If the authors would like to keep the reference model as a theoretically interesting method, it needs to be more comprehensive. To my knowledge, no such method exists in inference scaling literature. Similar methods might exist in speculative decoding literature to some extent, but this could itself be a paper on its own, showing the strengths and weaknesses of this type of reward modeling. Hence, I keep my current score.

---

### Meta-Review · Area_Chair_p1nd · 2025-12-14

**Summary:**

This work introduces Optimal Policy Minimum Bayesian Risk Decoding (OP-MBRD), a new inference-time decoding strategy for Large Language Models (LLMs). It builds upon minimum Bayes risk decoding by integrating reward and risk/similarity signals using a framework based on the optimal policy in KL-controlled reinforcement learning.
However, the paper received mixed reviews, leading to one marginal acceptance recommendation, one marginal rejection, and one clear rejection.

The main flaws identified during the review process and not yet addressed include:

(1) There lacks clarity in presentation and motivation. Reviewers consistently found the presentation confusing and disorganized. The core contribution and how OP-MBRD meaningfully differs from a simple twist on Best-of-N (BoN) with majority voting was a persistent point of contention and was not clearly established.

(2) There lacks additional baselines and experimental scope is limited. A reviewer suggested that the target audience is unclear, necessitating either more comprehensive results or a more self-contained theoretical presentation.

Given the above, I suggest a rejection and encourage the author to reorganize the manuscript.

**Reviewer Concerns:**

The main flaws identified during the review process and not yet addressed include:

(1) There lacks clarity in presentation and motivation. Reviewers consistently found the presentation confusing and disorganized. The core contribution and how OP-MBRD meaningfully differs from a simple twist on Best-of-N (BoN) with majority voting was a persistent point of contention and was not clearly established.

(2) There lacks additional baselines and experimental scope is limited. A reviewer suggested that the target audience is unclear, necessitating either more comprehensive results or a more self-contained theoretical presentation.

**Reviewer Scores:**

I hardly see the sign of review score increase for most of reviewers.

---

### Decision · Program_Chairs · 2026-01-26

Reject